# SoftMoE: Soft Differentiable Routing for Mixture-of-Experts in LLMs

**Mikołaj Zasada** [1]  **Łukasz Struski** [2]  **Jacek Tabor** [2 3]  **Marcin Kurdziel** [1]

## Abstract

Sparse Mixture-of-Experts (MoE) architectures enable scaling LLM parameters under a fixed inference budget by activating only a small subset of experts via top-$k$ routing. While this preserves causality and suits autoregressive language models, the discrete top-$k$ operator is not differentiable, forcing a fixed number of active experts per input and resulting in inefficient use of computation. We propose SoftMoE, which replaces discrete routing with a truncated soft top-$k$ LapSum relaxation, allowing gradient-based optimization of expert routing. We further parameterize the mean number of active experts per layer and impose a global budget constraint, enabling the model to learn how to allocate expert capacity across layers. SoftMoE remains fully compatible with autoregressive modeling and achieves performance comparable to or better than sparse MoE on language modeling and downstream tasks, while activating significantly fewer experts. Notably, the learned allocation is highly non-uniform, with later layers activating more experts. The source code is publicly available[†].

## 1. Introduction

The rapid progress of large language models has been driven primarily by scaling model capacity, which has been repeatedly shown to yield predictable improvements in performance (Kaplan et al., 2020; Hoffmann et al., 2022). While dense transformer architectures scale effectively, their inference cost grows linearly with the number of parameters, quickly becoming prohibitive. Mixture-of-Experts (MoE) architectures (Shazeer et al., 2017) address this limitation

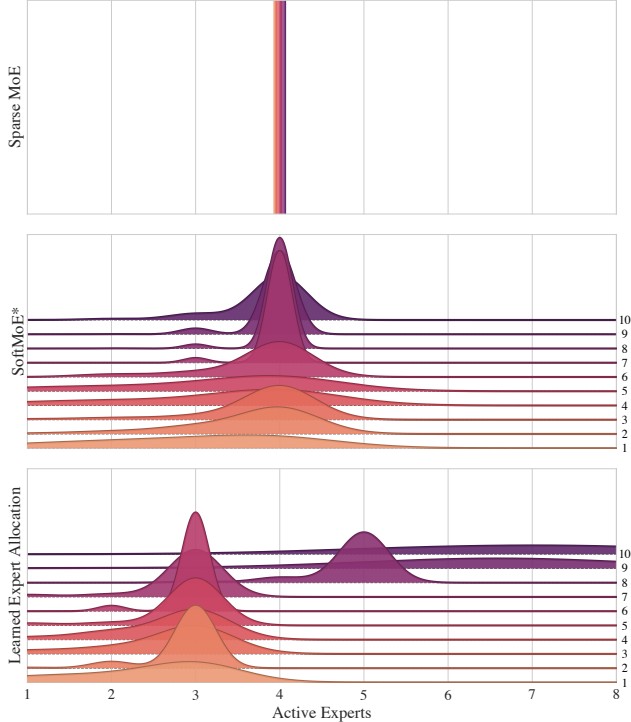

*Figure 1.* Learned expert allocation reveals non-uniform capacity needs across layers of a MoE model. Each color shows the distribution of activated experts in one layer of 10-layer model. Top: standard sparse MoE fixes the number of activated experts a priori. Middle: under soft top-$k$ routing, the number of activated experts depends on the input token. Bottom: learned allocation redistributes active experts budget: final layers activate more experts at the expense of initial and middle layers.

by enabling conditional computation: only a small subset of parameters is activated for each input token, allowing models to scale to hundreds of billions of parameters while keeping the per-token computation nearly constant.

In modern large-scale MoE models, sparse routing based on hard top-$k$ selection has emerged as the dominant paradigm (Fedus et al., 2022; Lepikhin et al., 2021). By activating only the $k$ highest-scoring experts per token, sparse MoE preserves autoregressive causality and achieves strong empirical performance. Yet the discrete nature of the top-$k$ operator introduces fundamental limitations. Since hard top-$k$ routing is non-differentiable, gradients cannot flow through the selection process, forcing the number of active

---
[1]AGH University of Krakow, Poland [2]Faculty of Mathematics and Computer Science, Jagiellonian University, Poland [3]Centre for Credible Artificial Intelligence, Warsaw University of Technology. Correspondence to: Mikołaj Zasada <mzasada@agh.edu.pl>.

*Proceedings of the $43^{rd}$ International Conference on Machine Learning*, Seoul, South Korea. PMLR 306, 2026. Copyright 2026 by the author(s).

[†]https://github.com/dlcuda/SoftMoE

experts to be fixed a priori. As a result, expert capacity cannot be adaptively allocated across layers or tokens, often leading to inefficient use of computation. Moreover, these models often exhibit unstable training dynamics (Fedus et al., 2022).

To address these limitations, several alternative routing strategies depart from the simple top-$k$ selection. For example, Puigcerver et al. (2024) replace discrete expert selection with differentiable token mixing, Zhou et al. (2022) improve capacity allocation by allowing experts to choose tokens, and Lewis et al. (2021) formulate routing as a linear assignment problem. While these approaches alleviate some of the limitations of top-$k$ routing, they often break compatibility with autoregressive modelling, decouple inference-time routing from the training procedure, or skew the inference-time expert load. This raises a question: can we design a routing mechanism that adaptively allocates expert capacity across layers and tokens, preserves causality and is computationally efficient?

In this work, we propose *SoftMoE*, a Mixture-of-Experts architecture built on a differentiable relaxation of the top-$k$ operator. Our approach replaces hard expert selection with a truncated soft top-$k$ mechanism based on the LapSum relaxation (Struski et al., 2025), enabling gradient-based optimization of expert routing. Crucially, unlike prior soft-routing methods, SoftMoE processes each token independently and remains fully compatible with autoregressive language modeling. In particular, SoftMoE does not require token mixing or expert-side token selection.

Our soft routing strategy also enables learning expert allocation across layers. To this end, SoftMoE explicitly parameterizes the mean number of active experts in each layer and impose a global constraint on the number of experts active across the network. This mechanism allows the model to learn how to redistribute computational capacity between layers under a fixed total inference cost. Specifically, layers that require more capacity can increase their expert allocation at the expense of layers that require fewer active experts. Such adaptive capacity redistribution is impossible under standard top-$k$ routing, which fixes the number of active experts a priori.

We evaluate SoftMoE on language modeling using C4 and OpenWebText, as well as downstream benchmarks. Our experiments demonstrate that SoftMoE consistently matches or outperforms standard sparse MoE while activating significantly fewer experts on average. Moreover, we observe that the learned capacity allocation is highly structured, with later layers consistently utilizing more experts. This provides new empirical insights into how large language models allocate conditional computation.

In summary, our contributions are threefold:

- We introduce a differentiable soft top-$k$ routing mechanism suitable for large-scale sparse MoE models.

- We propose a learnable, globally constrained expert budget that enables adaptive allocation of experts across layers.

- We demonstrate improved efficiency and competitive or superior performance on language modeling and downstream tasks, while maintaining autoregressive compatibility.

## 2. Related Work

Recent years have seen a gradual shift from monolithic transformer architectures toward Mixture-of-Experts (MoE) models and conditional computation (Jacobs et al., 1991; Jordan & Jacobs, 1994). The search for new architectures was initially motivated by empirical scaling laws observed in dense large language models (Hoffmann et al., 2022; Muennighoff et al., 2023), which demonstrated strong performance gains from increased model capacity. These laws were subsequently adapted to more compute-efficient sparse architectures (Ludziejewski et al., 2024; Clark et al., 2022), where dense layers are decomposed into expert modules and only a subset is activated per token. This conditional computation paradigm enables a substantial increase in model capacity while keeping the per-token computational cost nearly constant (Shazeer et al., 2017; Lepikhin et al., 2021). Empirical studies further demonstrate that sparse MoE models can outperform dense counterparts in machine translation and related sequence modeling tasks (Zhao et al., 2024). Concurrently, hybrid approaches that employ dense computation during training while retaining sparse execution at inference, such as DS-MoE (Pan et al., 2024), attempt to balance optimization stability with inference efficiency.

Despite these advantages, the effectiveness of MoE architectures is critically determined by the routing mechanism. Suboptimal token-to-expert assignment can result in load imbalance, degraded utilization of experts, and unstable training dynamics, which has motivated extensive work on routing simplification and stabilization strategies (Fedus et al., 2022; Zoph et al., 2022; Lewis et al., 2021). These challenges become more pronounced at scale, where routing design choices have been shown to significantly affect convergence and performance (Riquelme et al., 2021).

Historically, expert selection in MoE models has relied on hard top-$k$ routing, where each token is dispatched to the $k$ highest-scoring experts (Shazeer et al., 2017). While simple and effective, this discrete selection introduces non-differentiability, motivating a range of alternative formulations. Prior work explored expert-choice routing (Zhou et al., 2022), deterministic routing rules (Roller et al., 2021), global optimization approaches based on linear assignment

(Lewis et al., 2021) or optimal transport (Liu et al., 2023), as well as reinforcement learning–based routers (Bengio et al., 2015). However, these methods continue to rely on explicit discretization or indirect optimization and do not fully address the mismatch between training objectives and inference-time top-$k$ selection.

This mismatch has long been recognized in the broader context of top-$k$ learning. For example, direct top-$k$ error optimization was investigated as an alternative to top-1 optimization in SVMs (Lapin et al., 2016), with the central idea of aligning the training objective with inference-time selection by constructing differentiable loss functions that admit gradient-based optimization. This paradigm is relevant for MoE architectures, where routing decisions are inherently based on sparse top-$k$ selection. Subsequent work introduced smooth approximations of the top-$k$ operator that enable direct gradient optimization and often improve accuracy (Berrada et al., 2018). However, such approaches may incur computational overhead or trade accuracy for efficiency (Garcin et al., 2022). Optimal transport–based formulations offer smooth gradients via entropic regularisation (Cuturi et al., 2019; Xie et al., 2020; Masud et al., 2023), but their computational complexity may limit applicability to the large expert pools in modern MoE models.

Beyond MoE, top-$k$ learning has proven effective in sparse representation learning, sparse attention in computer vision, and recommender systems (Zhao et al., 2019; Chen et al., 2019; Hoefler et al., 2021; Chen et al., 2023), which share similarities with conditional computation. From a theoretical standpoint, top-$k$ optimization has also been linked to statistical learning theory, with recent results providing loss functions for optimizing cardinality of the predicted set (Cortes et al., 2024; Mao et al., 2024).

Motivated by limitations of hard top-$k$ expert selection, recent work has increasingly focused on soft routing mechanisms that relax discrete selection into continuous, fully differentiable operations. Puigcerver et al. (2024), for instance, propose to replace hard assignments with weighted combinations of tokens. Such mixing of tokens at different positions is, however, incompatible with autoregressive modelling. Antoniak et al. (2024) relies on batch inference and mixes tokens from different inputs. While this enables stable end-to-end optimization, mixing tokens from different inputs may potentially open the model to subtle cross-input attacks and information leaks. In this context, Struski et al. (2025) introduced a novel family of differentiable order-statistics operators, including soft ranking, soft top-$k$ selection, and soft permutations. Their method leverages a closed-form inverse of the LapSum function – defined as a sum of Laplace distributions – allowing low-memory, low-complexity selection of the highest activations. This makes the approach particularly well suited for large-scale MoE

routing, where scalability and efficiency are paramount.

## 3. Differentiable Top-$k$ Selection for Mixture of Experts

We focus on the construction of a *soft, fully differentiable* top-$k$ selection operator tailored to MoE architectures, following the framework introduced in Struski et al. (2025). In MoE models, top-$k$ selection plays a central role in expert routing, as it determines which subset of experts is activated for a given input. Consequently, the selection mechanism must both preserve sparsity and support efficient gradient-based optimization.

Let $\mathbf{r} = (r_i)_{i=1}^n \in \mathbb{R}^n$ denote the routing scores produced by a gating network over $n$ experts. The hard $\mathrm{Top_k}$ operator maps $\mathbf{r}$ to a binary routing vector:

$$\mathbf{p} = (p_i)_{i=1}^n \in \{0,1\}^n,$$

where $p_i = 1$ if expert $i$ belongs to the set of the $k$ highest-scoring experts and $p_i = 0$ otherwise. This operation enforces sparse expert activation and satisfies the cardinality constraint:

$$\sum_{i=1}^n p_i = k.$$

However, due to its discrete nature, hard $\mathrm{Top_k}$ routing is non-differentiable and typically requires surrogate gradients or heuristic approximations during training.

To overcome this limitation, we employ the differentiable relaxation of the $\mathrm{Top_k}$ operator (Struski et al., 2025) that replaces the binary routing vector with a continuous assignment:

$$\tilde{\mathbf{p}} = (\tilde{p}_i)_{i=1}^n \in [0,1]^n,$$

interpreted as soft expert selection weights. The relaxed operator preserves the expected sparsity pattern through the constraint:

$$\sum_{i=1}^n \tilde{p}_i = k,$$

for a (possibly non-integer) selection parameter $k$.

The construction formulates top-$k$ routing as a continuous order-statistics problem and introduces the *LapSum* function, defined as a sum of Laplace cumulative distribution functions. The method identifies a differentiable threshold $x$ by solving:

$$\mathrm{LapSum}(x) = \sum_{i=1}^n F_{\mathrm{Lap}}(r_i - x) = k,$$

where $r_i$ are the input scores and $F_{\mathrm{Lap}}$ denotes the Laplace CDF. This equation admits a unique solution $\tilde{x}$, which can be computed efficiently in closed form, without sorting or iterative optimization. As a result, the method achieves linear

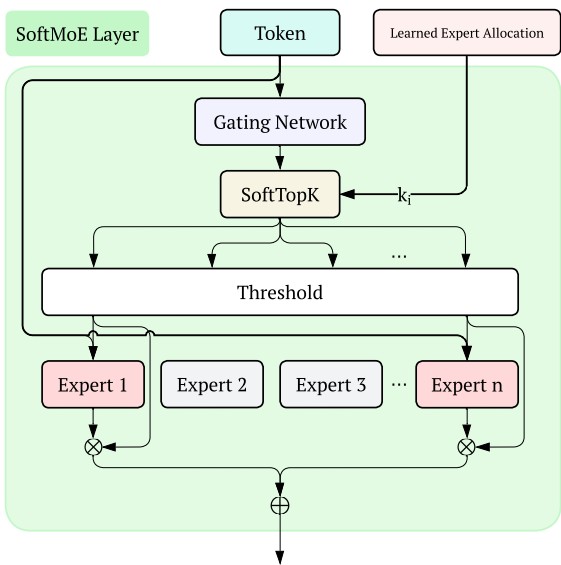

*Figure 2.* Architecture of a SoftMoE layer. The dense MLP layers in each transformer block are replaced by smaller expert MLPs with soft routing (light green box). The router computes scores over all experts, which are transformed by the SoftTopK operator into soft selection weights $p_i \in [0, 1]$. Weights below threshold are truncated to zero, yielding sparse but input-dependent expert activation.

time and memory complexity with respect to the number of experts, while remaining fully differentiable with respect to both the routing scores $r$ and the selection parameter $k$. We later exploit the latter property to optimize expert allocation patterns across MoE layers.

Intuitively, $\text{LapSum}(x)$ acts as a smooth approximation of the counting function $\sum_i \mathbf{1}[r_i \geq x]$, and solving $\text{LapSum}(x) = k$ corresponds to finding a continuous relaxation of the $k$-th order statistic (i.e., the top-$k$ threshold). The resulting soft selection weights are then given by $F_{\text{Lap}}(r_i - \tilde{x})$, which provide a differentiable approximation of the binary top-$k$ mask. This formulation can be viewed as a generalization of the softmax operator: similarly, it produces an output tensor of the same dimensionality as the input. However, unlike softmax (where the outputs form a probability distribution summing to 1), LapSum enforces that the outputs sum exactly to $k$, while each individual element remains in the range $[0, 1]$. This makes it particularly suitable for differentiable top-$k$ selection, where both sparsity and gradient flow are required.

## 4. From Discrete to Soft Expert Selection

In Section 3, we introduced a differentiable relaxation of the top-$k$ operator that maps routing scores $\mathbf{r} \in \mathbb{R}^n$ to soft selection weights $\tilde{\mathbf{p}} \in [0, 1]^n$, such that $\sum_1^n \tilde{p}_i = k$. We

now apply this relaxed $\text{SoftTopK}$ operator to Mixture-of-Experts routing.

Let $\mathbf{x} \in \mathbb{R}^d$ be an input token representation. In a standard sparse MoE architecture, the routing network selects which expert to activate for $\mathbf{x}$ using a discrete gating function:

$$G(\mathbf{x}) = \text{Top}_k\big(\text{softmax}(\mathbf{x}\mathbf{W}_g), k\big),$$

where $\mathbf{W}_g \in \mathbb{R}^{d \times n}$ is a learnable gating matrix and $n$ is the number of experts. The layer output is then calculated as a weighted combination of the experts' computation:

$$\mathbf{y} = \sum_{i=1}^{n} G(\mathbf{x})_i \cdot E_i(\mathbf{x}),$$

where $E_i : \mathbb{R}^d \to \mathbb{R}^d$ is the $i$-th expert network. Sparsity of the $\text{Top}_k$ operator guarantees that exactly $k$ experts contribute to the output. However, because $\text{Top}_k$ is non-differentiable, training gradient cannot flow through the expert selection process. Consequently, $\mathbf{W}_g$ is trained using the surrogate gradient from the pre-selection $\text{softmax}$ scores.

In SoftMoE (Figure 2), we replace the discrete $\text{Top}_k$ operator with its differentiable LapSum relaxation:

$$G(\mathbf{x}) = \mathcal{T}\big[\text{SoftTopK}(\mathbf{x}\mathbf{W}_g, k)\big],$$

where $\text{SoftTopK}(\cdot, k)$ is the soft top-$k$ operator introduced in Section 3. While the soft top-$k$ operator provides differentiability, naively routing through all $n$ experts using all soft weights would result in dense computation. Our goal, however, is to adapt the number of active experts to the input token and, subsequently, to learn how to allocate computational budget across layers. To recover computational efficiency, we therefore truncate the LapSum relaxation to remove low-contribution weights:

$$\mathcal{T}(z) = z\mathbb{I}_{z > \tau},$$

where $\mathbb{I}$ is the indicator function. For $z > \tau$, the gradient passes through the operator unchanged, while for $z < \tau$, the gradient is zero. At $z = \tau$ we adopt the subgradient convention $\mathcal{T}'(0) = 0$.

The truncation operator $\mathcal{T}(z)$ is equivalent to a shifted ReLU: active experts receive gradient through their routing weights, while inactive experts receive none. At the decision boundary, expert selection is non-differentiable, analogous to hard top-$k$ routing. That said, the model remains differentiable with respect to the selection parameter $k$ via the LapSum inverse. This gradient pathway is unavailable when $k$ is a fixed integer, and is what enables learning per-layer expert allocation under the global budget constraint. Moreover, because truncation applies a threshold rather than selecting by rank, the number of active experts

is input-dependent, allowing the model to adjust per-token compute. In summary, SoftMoE is a routing mechanism that enables gradient-based optimization of per-layer expert budgets and per-token compute adaptation, rather than a differentiable replacement of hard top-$k$ selection.

**Learning Expert Allocation Across Layers.**   A key advantage of the soft gating function is that the model becomes differentiable with respect to the expected number of active experts $k$. Consequently, we can *learn* how many experts to activate in each layer, rather than fixing this choice a priori. Such learned allocation of experts is impossible with standard sparse MoE architecture, which typically activates the same number of experts in each layer.

Consider a network with $L$ MoE layers. We parametrize expert allocation across layers with a vector $\mathbf{k} = (k_l)_{l=1}^{L}$, where $k_l$ denotes the mean number of active experts in layer $l$. Obviously, we require at least one active expert in each layer, i.e. $k_l \geq 1$, $l \in \{1, \ldots L\}$. Additionally, we need a constraint that prevents the network from learning to use all available experts, which would be equivalent to a dense computation. An intuitive way to achieve this is to introduce a global budget for active experts, and then let the optimization process allocate that budget across layers. We therefore optimize expert allocation under the following constraints:

$$
\begin{aligned}
k_l \geq 1, \quad l \in \{1, \ldots, L\} \\
\sum_{l=1}^{L} k_l = K
\end{aligned}
\quad , \quad (1)
$$

where $K$ is the global budget for active experts across all layers. This formulation ensures that the total computational cost remains controlled. With a budget of $K = 2L$, for instance, the average compute matches a sparse MoE model with $k = 2$ active experts per layer. That said, we expect learned allocation to match performance of a sparse MoE while using a tighter budget on active experts.

The constraints in Eq. 1 can be easily reformulated in an unconstrained domain. To this end, we introduce auxiliary parameters $\boldsymbol{\eta} = (\eta_l)_{l=1}^{L} \in \mathbb{R}^L$ and reparametrize expert allocation as:

$$
\boldsymbol{\pi} = \text{softmax}(\boldsymbol{\eta}), \quad k_l = \pi_l \cdot (K - L) + 1.
$$

Because $\text{softmax}$ ensures $\sum_l \pi_l = 1$, the affine transform maps the probability simplex to the feasible region where $\sum_l k_l = K$ and each $k_l \geq 1$. The reparametrization is end-to-end differentiable, enabling optimization of unconstrained allocation parameters $\boldsymbol{\eta}$ jointly with other model parameters, using standard gradient descent.

The budget constraint in Eq. 1 introduces competition between layers: increasing the expert allocation in one layer

requires matching compensation in other layers. Intuitively, this forces the model to redistribute computation to layers that benefit most from additional expert capacity. As we demonstrate in Section 5, this competition leads to highly non-uniform allocations. Importantly, learned expert allocation does not introduce any computational bottleneck: the available compute units can be utilized uniformly, by assigning units to layers proportionally to the learned means $k_l$.

Figure 3 illustrates the SoftMoE transformer architecture with learned allocation parameters.

**Computational Overhead.**   Operations in SoftMoE router add minimal overhead compared to sparse MoE routing. LapSum has $O(n)$ computational cost, where $n$ is the number of experts. Memory usage is also $O(n)$ per token vs. $O(k)$ for sparse MoE. Given typical expert counts in MoE layers ($n = 32$–$64$), this is negligible compared to the cost of multiplying by $n \times d$ gating matrix in both routers, and does not negate benefits of reduced average number of active experts.

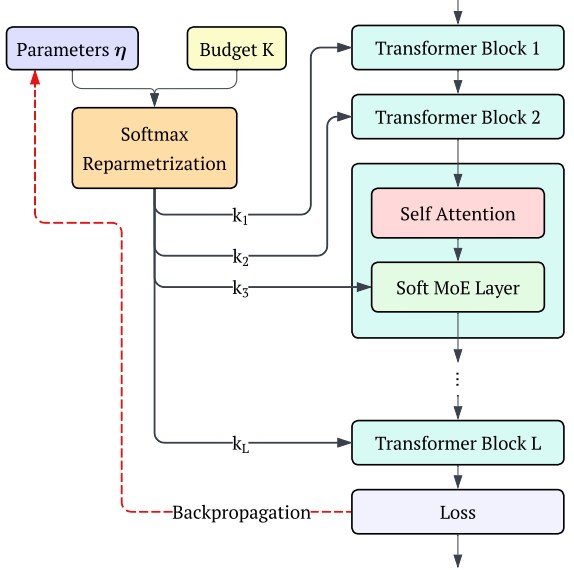

*Figure 3.* Transformer model with learned per-layer experts allocation and SoftMoE routing. Expert allocation is optimized with respect to unconstrained parameters $\boldsymbol{\eta}$. Softmax reparametrization ensures that each layer has at least one expert allocated ($k_l \geq 1$) and enforces global constraint on active experts ($\sum_l k_l = K$). Expert budget redistribution is end-to-end differentiable.

## 5. Experiments

We evaluate SoftMoE on autoregressive language modelling and downstream tasks. The experiments address two questions. First, whether SoftMoE routing matches or exceeds

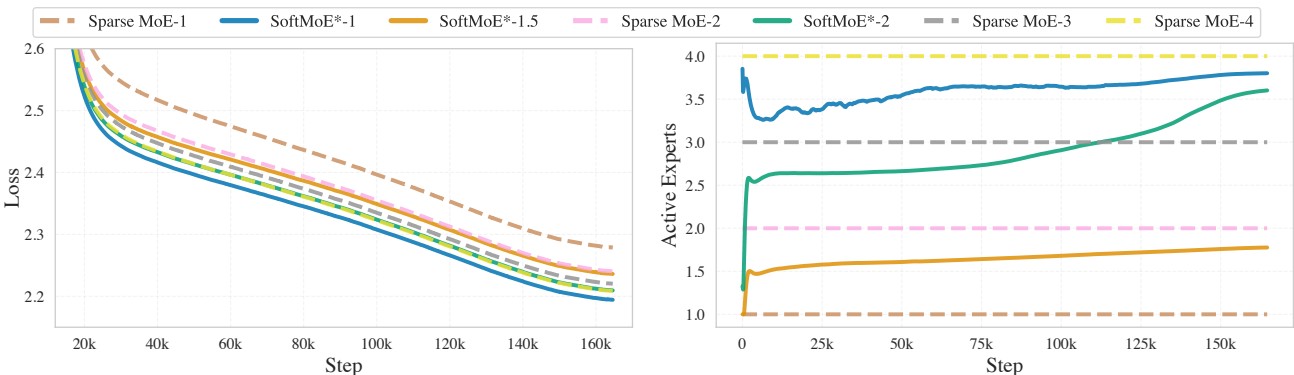

*Figure 4.* Training loss (left) and average number of active experts per token per layer (right) for Sparse MoE and SoftMoE* routing (no learned expert allocation across layers). Top-$k$ selection fixes the number of active experts a priori. Under soft routing, active experts count depends on the distribution of soft weights and evolves during training.

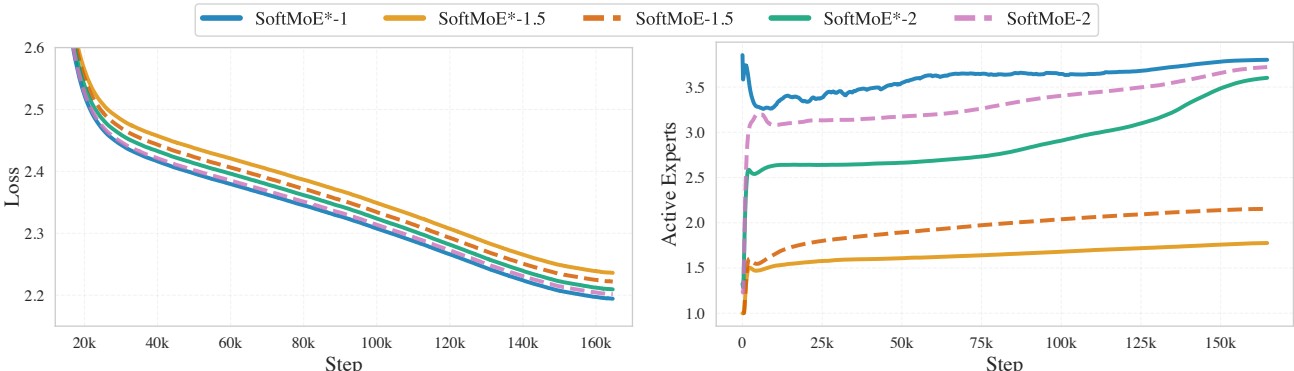

*Figure 5.* Training loss (left) and average number of active experts per token per layer (right) for soft routing without learning expert allocation (SoftMoE*) and with learned allocation (SoftMoE).

sparse MoE performance while using fewer active experts. Second, what allocation patterns emerge when the model is free to optimize expert allocation across layers under a fixed global budget. To answer these questions we compare SoftMoE against the Switch Transformer (Fedus et al., 2022) architecture, the de-facto standard MoE architecture for autoregressive language models. Except for different routing networks, all evaluated models share the same base architecture and pretraining protocol.

**Datasets and Downstream Tasks.** We train all models on two English text corpora of different scales: Open Web Text (OWT) (Gokaslan & Cohen, 2019) and Common Crawl (C4) (Raffel et al., 2020). For each corpus, we train a BPE tokenizer with a vocabulary of 32k tokens, yielding approximately 9B tokens for OWT and 206B tokens for C4. In each case we pretrain on 98% of the corpus and use the remaining 2% for validation.

After training, we evaluate each model on three zero-shot benchmarks: PIQA (Bisk et al., 2020) for physical commonsense reasoning, HellaSwag (Zellers et al., 2019) for sentence completion, and ARC-E (Bhakthavatsalam et al.,

2021) for elementary science questions. These tasks assess different reasoning capabilities and complement language modelling loss-based evaluation.

**Model Architecture and Training Protocol.** All evaluated models follow a decoder-only GPT-2 architecture (Radford et al., 2019), with dense MLP layers replaced by, respectively, Switch Transformer-based sparse MoE or SoftMoE layers. In each case we use a network with 10 transformer blocks and 32 experts per block. Each expert contains 5M parameters, yielding a total of 1.63B trainable parameters. We train standard sparse MoE models with $k \in \{1, 2, 3, 4\}$ active experts per layer, and SoftMoE models with initial mean number of active experts $k \in \{1, 1.5, 2\}$. We also train a variant of SoftMoE models with soft top-$k$ routing, but without learned expert allocation (SoftMoE*).

We train on C4 for 164k steps and on OWT for 18k steps. To maintain balanced expert load in each layer, all models are trained with auxiliary balancing loss (Fedus et al., 2022). Following standard practice, we also add small noise to the pre-activation values in MoE gating functions, encouraging networks to explore different routing patterns. All train-

ing runs were executed using Megatron-LM (Shoeybi et al., 2019) framework extended with SoftMoE routing. Complete training hyperparameters are provided in Appendix B.

**Implementation Details.** For computational efficiency, SoftMoE requires truncating low-contribution weights (Section 4). In practice, we set the truncation threshold $\tau$ by collecting activation statistics from initial input batches and adjusting the threshold so that initially the routing networks activate, on average, approximately $k$ experts per layer per token. Additionally, in each layer $i \in \{1, \ldots, L\}$ we upper-bound the number of active experts by $\lceil k_i \cdot \alpha \rceil$, where $k_i$ is the expected number of active experts and $\alpha \in \{2, 4\}$ is an expansion factor. This mechanism prevents OOM errors on tokens with near-uniform routing scores, while still allowing input-dependent expert selection to activate several-fold more experts than the learned mean.

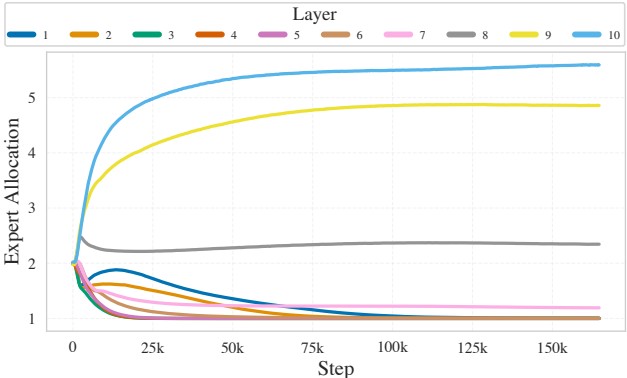

*Figure 6.* Evolution of the expert allocation parameters $k_l$ during training. SoftMoE architecture redistributes the active expert budget towards the final layers of the model.

### 5.1. Results

**SoftMoE\* vs. Sparse MoE.** We begin evaluation with comparison of sparse MoE baselines against SoftMoE routing without learning expert allocation (SoftMoE\* models). Table 1 reports results grouped by active expert budgets. Soft routing improves efficiency: across all configurations SoftMoE\* consistently achieves lower language modelling loss while activating fewer experts on average. The pattern is particularly evident with the $\leq 2$ experts per layer per token budget, where SoftMoE\* improves modelling loss while activating between $17\%$ (C4) and $24\%$ (OWT) fewer experts in training. Similar pattern is evident in inference, with SoftMoE\* consistently activating fewer experts on average.

This efficiency gain likely stems from differentiable routing. Specifically, SoftMoE\* can adapt the distribution of gating weights to the token representation, rather than committing to activate exactly $k$ experts per each input. In particular, given a simple token the routing network can concentrate

the weights on just one expert, whereas a token that benefits from additional capacity may be routed through a more uniform weight distribution, activating many experts. Indeed, we observe exactly this effect on empirical distributions of activated experts, especially in the initial layers of the model (Figure 1, middle plot).

Training dynamics for both architectures is shown in Figure 4. Unlike sparse MoE, which fixes active expert count a priori, SoftMoE\* adapts the number of activated experts during training, as the model learns which tokens benefit from more or less compute. This adaptation does not introduce any training instabilities.

**Learned Expert Allocation.** Next, we focus on SoftMoE models with learned expert allocation. Figure 6 reveals the typical allocation pattern learned on C4. Results for other configurations are reported in Appendix A. When given freedom to optimize expert allocation across layers, SoftMoE discovers markedly non-uniform allocation patterns. In particular, under a global constraint on active experts budget, SoftMoE redistributes expert capacity from early and middle layers towards final layers. Concretely, in our experiments the top 3 transformer layers absorb approximately $50\%$ of the total expert budget. This allocation pattern arises rapidly in initial stages of training, typically redistributing most of the budget within the first 25k training steps. The learned pattern translates to non-uniform expert allocation in inference (Figure 1, bottom plot).

The observed non-uniform allocation suggests that the model benefits from increased capacity in the final stages of token processing. This observation aligns with prior work on information processing in transformer models. For example (Tenney et al., 2019) used probing tasks to characterize representations across transformer network depth, and demonstrated that deeper layers of BERT capture high-level semantic information, while earlier layers encode simpler syntactic patterns. Related work by Jawahar et al. (2019) found that early layers of BERT models encode phrase-level information, while top layers encode semantic features. The learned expert allocation suggests similar phenomenon in decoder-only MoE models, with layers encoding semantic information benefiting from more expert capacity, and layers at initial stages of processing functioning effectively while activating fewer experts.

Our findings may have notable practical applications. Currently used MoE architectures usually adopt uniform expert allocation across layers. Our results, however, suggest that depth-aware allocation may be more appropriate.

Figure 5 compares training dynamics with (SoftMoE) and without (SoftMoE\*) learned expert allocation. Table 1 reports language modelling loss for both variants. Learned expert allocation typically improves language modelling

*Table 1.* Comparison of SoftMoE, SoftMoE with no learned expert allocation (SoftMoE*) and Sparse MoE models on OWT and C4 datasets. For each configuration, the table reports mean activated experts count during training (Train-AE) and inference (Infer-AE), mean activated parameters count (in millions) during inference (Active-Param), accuracy on the downstream tasks, validation loss and MoE layer configuration ($k$ and expansion factor $\alpha$). Results are grouped by the budget on average number of active experts per layer per token.

| Data | Model | $k$ | $\alpha$ | Train-AE | Infer-AE | Active-Param | PIQA | HELLA | ARC-E | Loss | #Expert |
|---|---|---|---|---|---|---|---|---|---|---|---|
| Open Web Text | Sparse MoE | 1 | - | 1 | 1 | 107.74 | 61.53 | 30.56 | 33.33 | 2.84 | $= 1$ |
| | Sparse MoE | 2 | - | 2 | 2 | 156.94 | 62.40 | 31.50 | 34.74 | 2.79 | $\leq 2$ |
| | SoftMoE* | 1.5 | 2 | 1.53 | 1.73 | 143.66 | **62.62** | **32.13** | **35.10** | **2.78** | $\leq 2$ |
| | SoftMoE | 1.5 | 2 | 1.63 | 1.96 | 154.98 | 62.57 | 32.12 | 35.45 | 2.76 | $\leq 2$ |
| | Sparse MoE | 3 | - | 3 | 3 | 206.14 | 63.55 | 32.16 | 34.92 | 2.77 | $\leq 3$ |
| | Sparse MoE | 4 | - | 4 | 4 | 255.34 | 63.87 | 32.36 | 36.68 | 2.75 | $\leq 4$ |
| | SoftMoE* | 1 | 4 | 3.64 | 3.73 | 242.06 | **63.93** | **33.68** | 36.51 | **2.70** | $\leq 4$ |
| | SoftMoE* | 2 | 2 | 2.62 | 3.12 | 212.05 | **63.82** | **32.50** | 37.21 | 2.75 | $\leq 4$ |
| | SoftMoE | 2 | 2 | 3.18 | 3.50 | 230.74 | 63.55 | 32.74 | **38.27** | 2.74 | $\leq 4$ |
| C4 | Sparse MoE | 1 | - | 1 | 1 | 107.74 | 69.75 | 43.14 | 42.5 | 2.27 | $= 1$ |
| | Sparse MoE | 2 | - | 2 | 2 | 156.94 | 71.98 | 45.41 | 40.74 | 2.24 | $\leq 2$ |
| | SoftMoE* | 1.5 | 2 | 1.65 | 1.81 | 147.60 | 71.06 | **45.49** | 39.15 | **2.23** | $\leq 2$ |
| | Sparse MoE | 3 | - | 3 | 3 | 206.14 | 71.60 | 46.36 | 42.50 | 2.22 | $\leq 3$ |
| | SoftMoE | 1.5 | 2 | 1.96 | 2.19 | 166.29 | 71.38 | **46.79** | **43.39** | **2.22** | $\leq 3$ |
| | Sparse MoE | 4 | - | 4 | 4 | 255.34 | 71.60 | 47.23 | 43.56 | 2.20 | $\leq 4$ |
| | SoftMoE* | 1 | 4 | 3.60 | 3.82 | 246.49 | **72.91** | **48.48** | 43.21 | **2.19** | $\leq 4$ |
| | SoftMoE* | 2 | 2 | 2.91 | 3.62 | 236.65 | **71.60** | **48.05** | 42.86 | 2.21 | $\leq 4$ |
| | SoftMoE | 2 | 2 | 3.35 | 3.76 | 243.54 | **72.03** | **47.48** | 42.15 | **2.20** | $\leq 4$ |

performance, though with a higher, on average, number of activated experts. This trade-off reflects the model's preference for concentrating additional capacity in later layers.

**Downstream Evaluation.** Downstream task performance confirms the benefits observed in language modelling loss (Table 1). On the largest HellaSwag benchmark, soft routing consistently outperforms Sparse MoE across all evaluated configurations on both pretraining datasets. Results for the PIQA benchmark follow a similar pattern, with soft routing achieving better or comparable accuracy in most configurations, while activating fewer experts. Importantly, configurations with the best pretraining loss tend to achieve strongest results on these two benchmarks, indicating that soft routing benefits translate to task-level performance. ARC-E exhibits higher variance across configurations, which we attribute to its smaller evaluation set. Nonetheless, SoftMoE achieves the strongest ARC-E performance under OWT pretraining and remains competitive under C4 pretraining, while activating fewer experts.

## 6. Conclusions

In this work, we presented SoftMoE, a Mixture-of-Experts architecture that replaces hard top-$k$ routing with a soft top-$k$ relaxation based on the LapSum operator. By addressing the non-differentiability inherent in standard sparse MoE routing, the proposed approach enables end-to-end gradient-based optimization of expert selection while maintaining sparsity, computational efficiency, and compatibility with autoregressive language modeling. This directly addresses the long-standing discrepancy between training-time optimization and inference-time routing decisions in MoE models.

A central property of SoftMoE is its ability to learn the allocation of expert capacity across layers under a fixed global computational budget. Rather than enforcing a predefined number of active experts per layer, the model is allowed to dynamically redistribute computation based on learned routing behavior. Empirical results indicate that this leads to highly non-uniform expert utilization patterns, with later layers consistently receiving a larger share of the computational budget. These observations provide additional insight into how conditional computation is exploited in deep language models.

Experimental evaluation on language modeling and downstream reasoning benchmarks shows that SoftMoE achieves performance comparable to or exceeding that of conventional sparse MoE architectures while activating fewer experts on average. This demonstrates that increased routing flexibility can yield improved compute efficiency without

degrading model quality.

The results suggest that differentiable top-$k$ routing constitutes a viable and scalable alternative to hard expert selection in MoE architectures. Beyond the specific LapSum-based relaxation introduced in this work, the proposed framework enables further investigation into adaptive conditional computation, including finer-grained budget control, interactions with scaling behavior, and extensions to larger-scale and multimodal models.

## Limitations

We evaluate our models on two English text corpora (OWT and C4) and English language downstream tasks. Evaluation of additional languages and tasks would provide a broader picture of SoftMoE capabilities. Another avenue for exploration not pursued in this work is applications of SoftMoE in multimodal models. Finally, our experiments are conducted on a 1.63B parameter architecture. While substantial, it is smaller than the largest language models deployed in practice. Our conclusions could, therefore, be strengthened by further validation at larger scales.

## Impact Statement

This paper presents work whose goal is to advance the field of machine learning. There are many potential societal consequences of our work, none of which we feel must be specifically highlighted here.

## Acknowledgments

This work was funded in whole or in part by the National Science Centre, Poland, and The National Centre for Research and Development, Poland, ARTIQ project: UMO-2021/01/2/ST6/00004 and ARTIQ/0004/2021, as well as by the "Excellence initiative–research university" program for AGH University of Krakow. Additionally, the work of Jacek Tabor and Łukasz Struski was supported by the National Science Centre, Poland, grants no. 2023/49/B/ST6/01137.

We gratefully acknowledge Polish high-performance computing infrastructure PLGrid (HPC Center: ACK Cyfronet AGH) for providing computer facilities and support within computational grant no. PLG/2025/018349.

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

# A. Learned expert allocation patterns

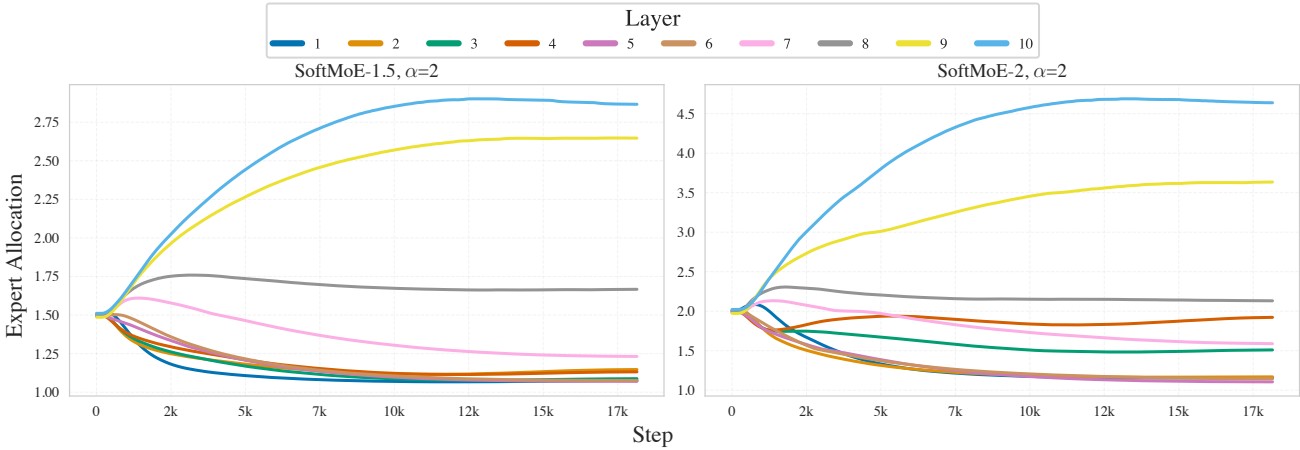

*Figure 7.* Evolution of the expert allocation parameters ($k_l$) during SoftMoE pretraining on OWT.

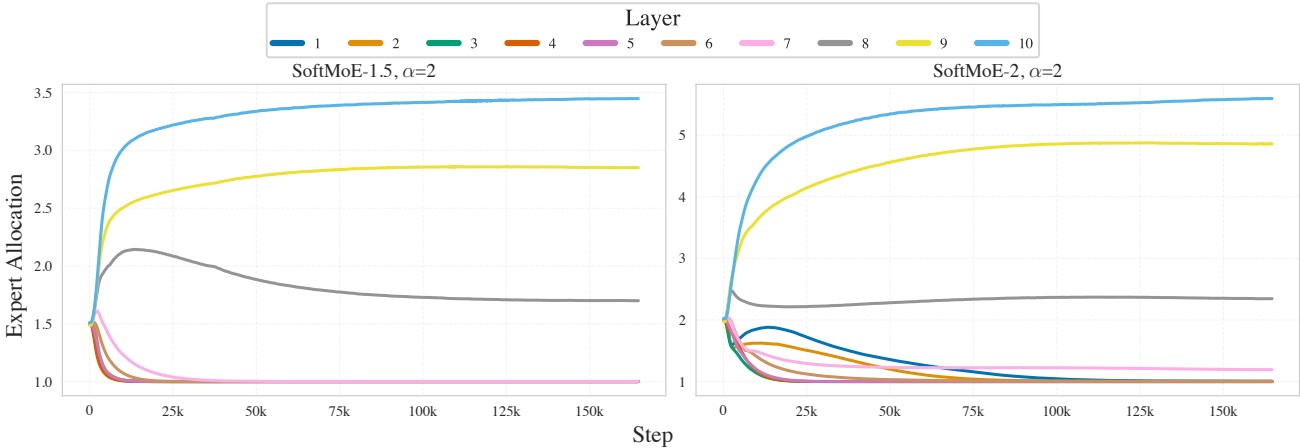

*Figure 8.* Evolution of the expert allocation parameters ($k_l$) during SoftMoE pretraining on C4.

# B. Training Hyperparameters

Configuration of models used in experiments is summarized in Table 2. All models were trained with mixed floating point precision (bfloat16 and float32). Experiments on the OWT dataset were conducted with a global batch size of 240, while those on the C4 dataset were conducted with a batch size of 600. We used eight GPUs to train the models on OWT and twenty GPUs to train on C4. In all configurations, we used a sequence length of 2048. The number of training steps depends on the dataset, and was 18k and 164k iterations for OWT and C4, respectively. We used the Adam optimizer with $\beta_1 = 0.9$ and $\beta_2 = 0.95$. The learning rate was annealed from $10^{-3}$ to $10^{-5}$ with a cosine decay. Additionally, we used a warm-up phase in the initial $1\%$ of training. All models contained 32 experts. An auxiliary loss function was used for intra-layer load-balancing. We also experimented with different jitter approaches in the standard sparse MoE architecture. However, they led to degraded performance (Table 8). Consequently, jitter was disabled in our sparse MoE baselines to ensure a fairer comparison, making the reported baselines stronger than configurations trained with jitter throughout. The LapSum-based SoftTopK implementation uses a temperature hyper-parameter that was annealed during training with a cosine decay from 5 to 1. We used RMSNorm in normalization layers and SiLU activation function. We trained our models with 8-way data parallelism for OWT and 20-way data parallelism for C4. Tensor parallelism and pipeline parallelism were not used. All experiments were computed on NVIDIA GH200 CPU-GPU chips with 96GB of VRAM and 120GB RAM each. Computing resources used in training are summarized in Table 3.

*Table 2.* Hyperparameters used to define model size.

| Model | Total Params | Blocks | Att. Heads | Experts | $d_{attn}$ | $d_{fnn}$ |
|-------|-------------|--------|-----------|---------|-----------|-----------|
| All | 1.631B | 10 | 10 | 32 | 640 | 2560 |

*Table 3.* Computing resources used for model training.

| Dataset | Model | Training Time (h) | Number of GPUs | GPUh |
|---------|-------|------------------|----------------|------|
| Open Web Text | Sparse MoE-1 | 3 | 8 | 23.7 |
| | Sparse MoE-2 | 2.2 | 8 | 17.7 |
| | Sparse MoE-3 | 2.3 | 8 | 18.5 |
| | Sparse MoE-4 | 2.6 | 8 | 20.6 |
| | SoftMoE*-1 | 3.2 | 8 | 25.7 |
| | SoftMoE*-1.5 | 2.5 | 8 | 20.0 |
| | SoftMoE-1.5 | 1.9 | 8 | 15.5 |
| | SoftMoE*-2 | 2.4 | 8 | 19.1 |
| | SoftMoE-2 | 2.4 | 8 | 19.4 |
| C4 | Sparse MoE-1 | 18.4 | 20 | 367.1 |
| | Sparse MoE-2 | 19.5 | 20 | 390.4 |
| | Sparse MoE-3 | 20.7 | 20 | 413.7 |
| | Sparse MoE-4 | 23.6 | 20 | 472.4 |
| | SoftMoE*-1 | 25.4 | 20 | 507.3 |
| | SoftMoE*-1.5 | 21.2 | 20 | 424.3 |
| | SoftMoE-1.5 | 23.8 | 20 | 475.0 |
| | SoftMoE*-2 | 24.7 | 20 | 493.5 |
| | SoftMoE-2 | 25.4 | 20 | 508.1 |

## C. Additional Results

*Table 4.* Comparison of the number of active experts, wall-clock training time, iteration time, throughput, and GPU utilization. The throughput and training times in these experiments are influenced by the computational cluster's load, as reflected in the GPU utilization.

| Dataset | Model | Train-AE | Time(h) | GPUh | Avg Iter Time(ms) | Tokens/s/gpu | GPU Util.(%) |
|---------|-------|----------|---------|------|-------------------|--------------|--------------|
| OWT | SparseMoE-2 | 2.00 | 2.2 | 17.7 | 438.96 | 142836 | 89.72 |
| | SoftMoE*-1.5 | 1.53 | 2.5 | 20.0 | 495.44 | 126554 | 82.75 |
| | SoftMoE-1.5 | 1.63 | 1.9 | 15.5 | 384.93 | 162883 | 94.02 |

*Table 5.* Relative variance in expert load (number of tokens per expert) in the SparseMoE and SoftMoE configurations. Lower values indicate a more balanced distribution of experts. In our experiments, SoftMoE shows no discernible impact on load balancing.

| Dataset | Model | $k$ | $\alpha$ | Load variance (relative) |
|---|---|---|---|---|
| OWT | SparseMoE | 1 | – | 0.10 |
| | SparseMoE | 2 | – | 0.05 |
| | SoftMoE* | 1.5 | 2 | 0.05 |
| | SoftMoE | 1.5 | 2 | 0.05 |
| | SparseMoE | 3 | – | 0.04 |
| | SparseMoE | 4 | – | 0.04 |
| | SoftMoE* | 1 | 4 | 0.03 |
| | SoftMoE* | 2 | 2 | 0.07 |
| | SoftMoE | 2 | 2 | 0.06 |
| C4 | SparseMoE | 1 | – | 0.04 |
| | SparseMoE | 2 | – | 0.04 |
| | SoftMoE* | 1.5 | 2 | 0.04 |
| | SparseMoE | 3 | – | 0.03 |
| | SoftMoE | 1.5 | 2 | 0.04 |
| | SparseMoE | 4 | – | 0.03 |
| | SoftMoE* | 1 | 4 | 0.02 |
| | SoftMoE* | 2 | 2 | 0.03 |
| | SoftMoE | 2 | 2 | 0.05 |

*Table 6.* Effect of truncation threshold $\tau$ on the downstream performance of SoftMoE* on the OWT dataset. The effective threshold is equal to $\tau \cdot k/E$, where $E$ is the number of experts and $k$ is the per-layer expert allocation.

| Configuration | $\tau$ | AE | PIQA | Hella | ARC-E |
|---|---|---|---|---|---|
| $k = 1, \alpha = 4$ | 1.5 | 2.47 | 62.35 | 32.78 | 36.51 |
| | 1.3 | 3.10 | 64.04 | 32.97 | 34.39 |
| | 1.1 | 3.73 | 63.93 | 33.68 | 36.51 |
| $k = 1.5, \alpha = 2$ | 1.8 | 1.73 | 62.63 | 32.13 | 35.10 |
| | 1.5 | 2.01 | 62.02 | 31.81 | 37.04 |
| | 1.3 | 2.35 | 63.87 | 32.16 | 37.04 |
| $k = 2, \alpha = 2$ | 1.8 | 1.81 | 62.57 | 31.34 | 35.27 |
| | 1.5 | 2.28 | 62.62 | 31.88 | 35.10 |
| | 1.3 | 2.70 | 62.57 | 32.54 | 35.10 |
| | 1.2 | 3.12 | 63.82 | 32.50 | 37.21 |

*Table 7.* Ablation of truncation effects during inference-time. Results are reported for SoftMoE-2 trained with fixed $\tau = 1.2$ and evaluated across different truncation levels in inference. The effective threshold is equal to $\tau \cdot k_l / E$, where $E$ is the number of experts and $k_l$ is the learned expert allocation for layer $l$.

| Dataset | $\tau$ | AE | PIQA | Hella | ARC-E |
|---------|--------|------|-------|-------|-------|
| OWT | 2.4 | 1.71 | 62.94 | 32.33 | 36.68 |
|     | 1.8 | 2.21 | 63.49 | 32.82 | 37.56 |
|     | 1.2 | 3.50 | 63.55 | 32.74 | 38.27 |
|     | 0.6 | 4.52 | 63.65 | 32.69 | 37.21 |
| C4  | 2.4 | 1.62 | 71.00 | 46.98 | 40.03 |
|     | 1.8 | 1.68 | 71.65 | 47.56 | 41.44 |
|     | 1.2 | 3.75 | 72.03 | 47.48 | 42.15 |
|     | 0.6 | 4.69 | 71.92 | 47.47 | 41.62 |

*Table 8.* Effects of the routing jitter in sparse MoE baselines. In most cases, jitter degrades baseline performance.

| Jitter | Model | PIQA | Hella | ARC-E | Loss |
|--------|-------|-------|-------|-------|------|
| No | Top-1 | 60.77 | 28.79 | 36.86 | 3.06 |
| No | Top-2 | 61.20 | 30.02 | 36.50 | 2.99 |
| No | Top-3 | 61.86 | 30.02 | 34.39 | 2.97 |
| 0.05 | Top-1 | 60.44 | 29.21 | 34.92 | 3.05 |
| 0.05 | Top-2 | 60.72 | 29.08 | 33.86 | 3.00 |
| 0.05 | Top-3 | 61.26 | 30.12 | 35.27 | 2.97 |

# D. Contributions

Mikołaj was responsible for method development, implementation of both the proof-of-concept and final versions of the SoftMoE model, execution of experiments, performance evaluation, development of the research infrastructure and datasets preparation. Marcin proposed the initial research concept, contributed to algorithm development, assisted in analysing results and suggesting experiments, and coordinated the project. Łukasz was responsible for implementing the soft top-$k$ algorithm, supported its understanding and adaptation in the system, and made contributions to the development of the method. Jacek addressed theoretical aspects of the work, proposed unconstrained parametrization of the budget allocator, and contributed to research direction and algorithm development. All authors contributed to the manuscript.

