# OpenReview forum: "SoftMoE: Soft Differentiable Routing for Mixture-of-Experts in LLMs"
_ICML.cc/2026/Conference — ICML 2026 regular_

### Official Review · Reviewer_7XHP · 2026-03-11

**Soundness:** 3
**Presentation:** 3
**Significance:** 3
**Originality:** 3
**Overall Recommendation:** 5
**Confidence:** 3

**Summary:**

This paper proposes SoftMoE, a variant of sparse Mixture-of-Experts that replaces the non-differentiable top-k routing with a truncated soft top-k LapSum relaxation. This design enables gradient-based optimization of expert routing and allows the model to learn the average number of active experts per layer under a global compute budget. Experiments show that SoftMoE achieves comparable or better performance than standard sparse MoE on language modeling and downstream tasks while activating fewer experts. The learned allocation also reveals a non-uniform pattern, where later layers tend to activate more experts.

**Compliance With Llm Reviewing Policy:**

Affirmed.

**Final Justification:**

The rebuttal addressed my main concerns, so I maintain my score of 5.

**Key Questions For Authors:**

See weaknesses.

**Limitations:**

yes

**Strengths And Weaknesses:**

**Strengths**

1. The ideas of SoftTopk, which transform expert selection into a differentiable operation, and the introduction of a global budget for expert allocation across layers are novel and interesting.

2. The work is well-motivated, and the proposed method is practical.

3. The experiments are well-designed and comprehensive, covering both training and inference performance.

**Weaknesses**

1. The paper lacks a detailed description of LapSum, despite citing a related paper, even though it is a crucial component of the proposed method.

2. The paper lacks analysis and ablation studies on the parameter $\tau$, which plays an important role in expert selection.

---

> ### Author Rebuttal · Authors · 2026-03-28
>
> Thank you Reviewer for the careful reading of our paper and constructive feedback. We address each raised concern below.
>
> > W1: LapSum formula
>
> To clarify the soft top-$k$ selection mechanism, we will add the explicit formulation to Section 3 in the paper. The method identifies a differentiable threshold $x$ by solving the equation:
>
> $$
> \mathrm{LapSum}(x) = \sum_{i=1}^n F_{\text{Lap}}(r_i - x) = k,
> $$
>
> where $r_i$ are the input scores and $F_{\text{Lap}}$ denotes the Laplace CDF.
> This equation admits a unique solution $\tilde{x}$, which we denote as $\tilde{x} = \mathrm{LapSum}^{-1}(k)$ and is described by an explicit formula. Intuitively, $\mathrm{LapSum}(x)$ acts as a smooth approximation of the counting function $\sum_i \mathbf{1}[r_i \ge x]$, and solving $\mathrm{LapSum}(x)=k$ corresponds to finding a continuous relaxation of the $k$-th order statistic (i.e., the top-$k$ threshold). The resulting soft selection weights are then given by $F_{\text{Lap}}(r_i - \tilde{x})$, which provide a differentiable approximation of the binary top-$k$ mask. This formulation can be viewed as a generalization of the softmax operator: similarly, it produces an output tensor of the same dimensionality as the input. However, unlike softmax (where the outputs form a probability distribution summing to 1), LapSum enforces that the outputs sum exactly to $k$, while each individual element remains in the range $[0, 1]$. This makes it particularly suitable for differentiable top-$k$ selection, where both sparsity and gradient flow are required.
>
> > W2: Ablation of $\tau$
>
> Below we report how changing truncation threshold affects downstream performance ($E$ - number of experts, $k_l$ - learned experts allocation, $k$ - per-layer allocation).
>
> 1) $\tau$ at training and inference. OWT. SoftMoE*. Effective threshold: $\tau \cdot k / E$
>
> |Model|τ|AE|PIQA|Hella|ARC-E|
> |-|-:|-:|-:|-:|-:|
> |k=1, α=4|1.5|2.47|62.35|32.78|36.51|
> |k=1, α=4|1.3|3.10|64.04|32.97|34.39|
> |k=1, α=4|1.1|3.73|63.93|33.68|36.51|
> |k=1.5, α=2|1.8|1.73|62.63|32.13|35.10|
> |k=1.5, α=2|1.5|2.01|62.02|31.81|37.04|
> |k=1.5, α=2|1.3|2.35|63.87|32.16|37.04|
> |k=2, α=2|1.8|1.81|62.57|31.34|35.27|
> |k=2, α=2|1.5|2.28|62.62|31.88|35.10|
> |k=2, α=2|1.3|2.70|62.57|32.54|35.10|
> |k=2, α=2|1.2|3.12|63.82|32.50|37.21|
>
> 2) $\tau$ at inference. SoftMoE-2 trained at $\tau = 1.2$. Effective threshold: $\tau \cdot k_l / E$
>
> |Dataset|τ|AE|PIQA|Hella|ARC-E|
> |-|-:|-:|-:|-:|-:|
> |OWT|2.4|1.71|62.94|32.33|36.68|
> |OWT|1.8|2.21|63.49|32.82|37.56|
> |OWT|1.2|3.50|63.55|32.74|38.27|
> |OWT|0.6|4.52|63.65|32.69|37.21|
> |C4|2.4|1.62|71.00|46.98|40.03|
> |C4|1.8|1.68|71.65|47.56|41.44|
> |C4|1.2|3.75|72.03|47.48|42.15|
> |C4|0.6|4.69|71.92|47.47|41.62|

---

> > ### Author Rebuttal · Reviewer_7XHP · 2026-04-02
> >
> > The authors provide additional explanation and ablation studies. I **maintain my score of 5**.

---

> > > ### Author Response · Authors · 2026-04-02
> > >
> > > We appreciate that the additional explanations provided were sufficient, and we are glad to hear that our responses have been satisfactory. Thank you as well for your valuable feedback and positive response throughout this process.

---

### Official Review · Reviewer_rwe7 · 2026-03-13

**Soundness:** 3
**Presentation:** 4
**Significance:** 3
**Originality:** 3
**Overall Recommendation:** 4
**Confidence:** 4

**Summary:**

This paper investigates the non-differentiability issue of hard top-k routing in sparse MoE. The authors propose SoftMoE: replacing discrete top-k with a soft top-k based on LapSum, and restoring sparse execution through threshold truncation. Building on this, they introduce a parameterization of inter-layer expert budgets that satisfies a global constraint ($\sum_l k_l = K$), enabling the learning of average expert allocation per layer. The experiments utilize a 1.63B parameter, 10-transformer-block, decoder-only GPT-2 style model with 32 experts per layer, pre-trained on OWT/C4 and evaluated on PIQA, HellaSwag, and ARC-E. In terms of literature positioning, this work serves more as an integration of existing differentiable top-k theoretical tools into mainstream token-wise, autoregressive-compatible MoE routing, rather than proposing a fundamentally new MoE theoretical framework.

**Compliance With Llm Reviewing Policy:**

Affirmed.

**Final Justification:**

See rebuttal ack.

**Key Questions For Authors:**

1. Can you provide comparisons that are exactly compute-matched and latency-matched, rather than just "$\le 2 / \le 4$" grouped budget comparisons? Table 1 proves competitiveness at a lower average AE but does not strictly prove superiority at the same end-to-end cost.
2. Can you systematically ablate $\tau, \alpha$, temperature annealing, routing jitter, and auxiliary load-balancing loss? Specifically, does the concentration of budget in later layers change significantly with these implementation choices?

**Limitations:**

The paper acknowledges three limitations: evaluation only on English OWT/C4 and English downstream tasks, model scale limited to 1.63B, and no study of multimodal settings. I agree these are substantive.

**Strengths And Weaknesses:**

Strengths
1. Highly relevant topic aligned with mainstream MoE pain points. Both Switch Transformer and ST-MoE identify routing design, training stability, and expert utilization as core challenges; this paper targets one of the most critical bottlenecks in current sparse LLMs.
2. Clear design boundaries. SoftMoE maintains token-wise routing without token mixing or switching to expert-side token selection. Simultaneously, it implements globally constrained inter-layer budget learning via softmax reparameterization on a simplex, resulting in a concise form that is engineering-friendly for integration into existing routed-LLM frameworks.
3. Direct experimental evidence for the efficacy of soft routing. The authors provide a comparison between SoftMoE* and sparse MoE using consistent backbones, training protocols, and data, making the attribution to the router itself clearer than in many MoE papers.
4. Valuable observations on non-uniform inter-layer expert allocation. Figures 6, 7, and 8 show budgets quickly concentrating in later layers. This is an empirical phenomenon worthy of future mechanistic study, even if a rigorous explanation is not yet provided.

Weaknesses

1. Limited novelty boundaries. The differentiable top-k core of SoftMoE is not an original contribution of this paper but is built upon LapSum[1]. More broadly, differentiable/sparse gating has previously been explored in DSelect-k[2], and fully differentiable MoE for autoregressive LM pre-training already exists in Lory[3]. Therefore, the core novelty is more accurately described as the "systematic integration of LapSum into token-wise autoregressive MoE + learnable layer budgets" rather than a brand-new theory.
2. Insufficient proof that learned layer allocation is an independent source of gain. Within the same nominal budget group, SoftMoE often exhibits both lower loss and higher actual AE compared to SoftMoE. For instance, SoftMoE-1.5 on OWT outperforms SoftMoE-1.5 in loss but also has a higher Train-AE (1.63 vs. 1.53). Thus, improvements may partially stem from increased realized compute rather than just optimized inter-layer allocation.
3. Gaps in fairness and attribution. SoftMoE uses a routing jitter of $10^{-2}$ during training, which the paper explicitly notes causes performance degradation in standard sparse MoE and was thus disabled for the baseline. Combined with the lack of systematic ablations for $\tau, \alpha$, temperature annealing, and auxiliary load-balancing loss, current experiments are insufficient to attribute gains solely to the "soft routing mechanism." Recent work on load-balancing implementations also indicates such choices substantially 4. Lack of system-level evidence for "minimal overhead / no bottleneck." The paper does not report latency, throughput, expert-parallel communication, or kernel efficiency, providing only GPUh. Table 3 shows that on C4, the GPUh for several SoftMoE configurations is actually higher than sparse baselines. At present, one can only claim that the additional routing overhead does not render the method uncompetitive, rather than making strong claims about deployment efficiency.

[1] Struski, Ł., Bednarczyk, M. B., Podolak, I. T., & Tabor, J. (2025). LapSum--One Method to Differentiate Them All: Ranking, Sorting and Top-k Selection. arXiv preprint arXiv:2503.06242.

[2] Hazimeh, H., Zhao, Z., Chowdhery, A., Sathiamoorthy, M., Chen, Y., Mazumder, R., ... & Chi, E. (2021). Dselect-k: Differentiable selection in the mixture of experts with applications to multi-task learning. Advances in Neural Information Processing Systems, 34, 29335-29347.

[3] Zhong, Z., Xia, M., Chen, D., & Lewis, M. Lory: Fully differentiable mixture-of-experts for autoregressive language model pre-training. 2024. URL https://api. semanticscholar. org/CorpusID, 268891288.

---

> ### Author Rebuttal · Authors · 2026-03-27
>
> Thank you for the thorough and technically detailed review.
>
> > W1: Limited novelty boundaries
>
> DSelect-k has not been applied to autoregressive language modeling. It is also unclear how it could be applied at scale: DSelect-k activates all experts during training, converging to sparse selection only asymptotically via entropy regularization. In language modeling this would amount to dense LLM computation throughout most of training.
>
> Lory replaces sparse routing with soft averaging of expert parameters into a single merged FFN. Unlike sparse routing, this introduces parallelism constraints: expert-parallel deployment requires either all-to-all averaging of parameters or additional all-to-all communication of hidden activations. In practice, during inference Lory makes a single routing decision from the prompt's average hidden representation. The resulting merged FFN is fixed for the entire generation.
>
> Neither method supports gradients through $k$ or learned inter-layer allocation. In contrast, the contribution of our work is that we not only introduce efficient differentiable routing, but also push differentiability further by enabling adaptive, globally-constrained expert allocation across layers. While we do not compare directly against Lory, we provide a comparison with Mixture of Tokens (MoT) [arxiv:2310.15961], another differentiable, per-token method that mixes tokens from from different inputs in the batch (table bellow). We outperform MoT while activating fewer parameters.
>
> | Data | Model | Active-Param | PIQA  | HELLA | ARC-E |
> | - | - | -:| -:| -:| -:|
> | OWT (9.1B) | SoftMoE*-1 | 242 | 63.93 | 33.68 | 36.51 |
> | OWT (9.1B) | SoftMoE*-2 | 212 | 63.82 | 32.50 | 37.21 |
> | OWT (9.1B) | SoftMoE-2  | 231 | 63.55 | 32.74 | 38.27 |
> | 5% C4 (10.3B) | MoT 32E/1 | 336 | 62.40 | 31.10 | 37.30 |
>
> > W2: Insufficient proof that learned layer allocation is an independent source of gain
>
> We acknowledge that learned allocation tends to increase the realized active expert count. However, we observe that, on average, our implementations outperform the baseline. More importantly, the model does not increase AE uniformly: it redistributes compute budget toward later layers, converging to the same non-uniform pattern across our configurations and datasets. A more uniform distribution would support the hypothesis that gains stem from additional compute alone. The consistent structural reallocation suggests, however, that the mechanism is making a meaningful architectural decision. We agree with the view that this non-uniform allocation is an interesting empirical phenomenon worthy of further study.
>
> > W3: Gaps in fairness and attribution
>
> We ran sparse MoE baselines with standard jitter for hard top-{1,2,3} configurations (table bellow). Where jitter had an effect on hard top-$k$, it typically degraded the baseline. We therefore disabled jitter for sparse MoE to provide a fairer comparison. In particular, the reported baselines are stronger than if we decided to use jitter across the board.
>
> | Jitter | Model | PIQA  | Hella | ARC-E | Loss |
> |-|-|-:|-:|-:|-:|
> | No     | Top-1 | 60.77 | 28.79 | 36.86 | 3.06 |
> | No     | Top-2 | 61.2  | 30.02 | 36.5  | 2.99 |
> | No     | Top-3 | 61.86 | 30.02 | 34.39 | 2.97 |
> | 0.05   | Top-1 | 60.44 | 29.21 | 34.92 | 3.05 |
> | 0.05   | Top-2 | 60.72 | 29.08 | 33.86 | 3.00 |
> | 0.05   | Top-3 | 61.26 | 30.12 | 35.27 | 2.97 |
>
> > Q1: Compute-matched comparison
>
> In the table bellow we report active experts, wall clock training time, time per iteration, throughput and GPU utilization. Throughput and timings in those experiments are affected by compute cluster load, which is not under our control (manifested in GPU utilization). Thats said, the baselines are scaled to be computed matched with SoftMoE taking into account theoritical flops (active experts). Note that except for the number of active experts, all compared models use the same architecture.
>
> | Dataset | Model        | Train-AE | Time(h) | GPUh | Avg Iter Time(ms) | Tokens/s/gpu | GPU Util.(%) |
> | ------- | ------------ | -------: | ------: | ---: | ----------------: | -----------: | -----------: |
> | OWT     | SparseMoE-2  |        2 |     2.2 | 17.7 |            438.96 |       142836 |        89.72 |
> | OWT     | SoftMoE*-1.5 |     1.53 |     2.5 | 20.0 |            495.44 |       126554 |        82.75 |
> | OWT     | SoftMoE-1.5  |     1.63 |     1.9 | 15.5 |            384.93 |       162883 |        94.02 |
>
> > Q2 and W4
>
> Regarding ablations and the concentration of budget in later layers: in our sweeps across hyper-parameter values, large training runs consistently displayed concentration of expert budget in later layers (Figs. 6–8 in the paper).

---

> > ### Author Rebuttal · Reviewer_rwe7 · 2026-04-04
> >
> > Good luck. I will raise my score from 3 to 4.

---

> > > ### Author Response · Authors · 2026-04-04
> > >
> > > Thank you for your thorough evaluation and overall assessment of our paper. We are glad that the rebuttal resolved your concerns.

---

### Official Review · Reviewer_Fdmu · 2026-03-13

**Soundness:** 2
**Presentation:** 3
**Significance:** 2
**Originality:** 2
**Overall Recommendation:** 4
**Confidence:** 4

**Summary:**

This paper proposes SoftMoE, a Mixture-of-Experts routing method that replaces standard hard top-k routing with a LapSum-based soft top-k relaxation. The method also introduces a learnable globally constrained per-layer expert budget. Experiments on GPT-2-style MoE models trained on C4 and OWT show competitive or improved language modeling and downstream performance.

**Compliance With Llm Reviewing Policy:**

Affirmed.

**Final Justification:**

My original main concern is on the claimed differentiability of the proposed method, where I believe the truncation make SoftMoE still non-differentiable on expert selection boundary. The rebuttal and follow-up clarify this and emphasize the differentiability on the number of experts as one of the main advantage. My other concerns on experiments and LapSum formula were addressed by the rebuttal.
Therefore, I raised my score from 3 to 4.

**Key Questions For Authors:**

Included above.

**Limitations:**

Included above.

**Strengths And Weaknesses:**

### Strength

* The paper aims to addresses meaningful MoE design questions: differentiability of router and adaptive expert allocation.

* The observed learned allocation patterns across layers are interesting, e.g. later layers tend to receive more expert capacity.

### Weakness

* The claimed differentiability benefit is only partial in the proposed router. The paper motivates from replacing non-differentiable top-k with a differentiable LapSum relaxation, but the deployed router is still using hard truncation. After which the gradients are still zero for experts below threshold, and cannot provide much valuable info for the learning of the gate.

* Aside from the softtopk, the method introduces two separate mechanisms: truncation makes the number of active experts input-dependent at the token level, while a global-budget parameterization learns per-layer mean expert counts. These two controls appear to overlap, and the experiment does not analyze their interaction.

* TopK can also be seen as a truncation. Operationally, the proposed method replaces Topk(softmax(xW_g), k) with Truncate(SoftTopK(xW_g,k)). This raises the question of whether the gains come specifically from the LapSum relaxation, or more generally from using soft weights plus thresholding. It would be helpful to compare against a simpler baseline like thresholded softmax routing. Also, in the SoftMoE formula, softmax is removed, could the author explain the reason?

* The paper do not include the key formulation of the lapsum. It would be necesssary to specifically include the formula in the main text for better understanding.

---

> ### Author Rebuttal · Authors · 2026-03-28
>
> Thank you Reviewer for the careful reading of our paper and constructive feedback. We address each raised concern below.
> > W1: Differentiability is only partial
>
> The truncation operator $\mathcal{T}(z) = z\mathrm{I}_{z>\tau}$ is in essence equivalent to a shifted ReLU activation. In this sense it follows standard practice in deep learning: sparse activations all zero out subsets of outputs, yet preserve useful gradient flow. Additionally, the truncated experts are those with small soft weights. Since gradient contributions scale multiplicatively with weight magnitude, the zeroed terms carry minor signal. In contrast, hard top-k zeros gradients for all but exactly $k$ experts, regardless of score distribution. Finally, SoftMoE remains differentiable with respect to the selection parameter $k$ through the LapSum inverse, enabling learned per-layer allocation. No such gradient pathway is available in hard top-k or alternative soft-weighting schemes. We also note that auxiliary load balancing, which we use with SoftMoE, ensures no expert is systematically truncated across the training distribution.
> > W2: Truncation and global-budget overlap
>
> Our experiments isolate the contributions of truncation and learned allocation to the extent that is practically meaningful. In particular, SoftMoE* uses soft routing with truncation, but uniform $k$ across layers. Full SoftMoE adds learned allocation on top of SoftMoE*. Table 1 in the paper presents both variants side by side. We do not ablate learned allocation without truncation, as this would activate all $n$ experts per token (equivalent to dense computation), which defeats the purpose of sparsely activated MoE architecture.
>
> Below we report how changing truncation threshold affects downstream performance ($E$ - number of experts, $k_l$ - learned experts allocation, $k$ - per-layer allocation).
> 1) $\tau$ at training and inference. OWT. SoftMoE*. Eff. th.: $\tau \cdot k / E$
> ||τ|AE|PIQA|Hella|ARC-E|
> |-|-:|-:|-:|-:|-:|
> |k=1, α=4|1.5|2.47|62.35|32.78|36.51|
> |k=1, α=4|1.3|3.10|64.04|32.97|34.39|
> |k=1, α=4|1.1|3.73|63.93|33.68|36.51|
> |k=1.5, α=2|1.8|1.73|62.63|32.13|35.10|
> |k=1.5, α=2|1.5|2.01|62.02|31.81|37.04|
> |k=1.5, α=2|1.3|2.35|63.87|32.16|37.04|
> |k=2, α=2|1.8|1.81|62.57|31.34|35.27|
> |k=2, α=2|1.5|2.28|62.62|31.88|35.10|
> |k=2, α=2|1.3|2.70|62.57|32.54|35.10|
> |k=2, α=2|1.2|3.12|63.82|32.50|37.21|
> 2) $\tau$ at inference. SoftMoE-2 trained at $\tau = 1.2$. Eff. th.: $\tau \cdot k_l / E$
> ||τ|AE|PIQA|Hella|ARC-E|
> |-|-:|-:|-:|-:|-:|
> |OWT|2.4|1.71|62.94|32.33|36.68|
> |OWT|1.8|2.21|63.49|32.82|37.56|
> |OWT|1.2|3.50|63.55|32.74|38.27|
> |OWT|0.6|4.52|63.65|32.69|37.21|
> |C4|2.4|1.62|71.00|46.98|40.03|
> |C4|1.8|1.68|71.65|47.56|41.44|
> |C4|1.2|3.75|72.03|47.48|42.15|
> |C4|0.6|4.69|71.92|47.47|41.62|
>
> > W3: Gains from LapSum specifically vs soft weights + thresholding
>
> The key property that distinguishes LapSum from simpler alternatives such as thresholded softmax is differentiability with respect to the selection parameter $k$. A thresholded softmax can produce variable expert counts per token, but there is no gradient pathway to learn expert allocation across layers. LapSum provides this gradient through its closed-form inverse, which is what enables the learned per-layer budget (Section 4 in the paper).
>
> Regarding softmax removal: SoftTopK maps logits to $\tilde{\mathbf{p}} \in [0,1]^n$ with $\sum_i \tilde{p}_i = k$. It therefore already outputs soft selection weights in the target domain. Applying softmax beforehand would double-normalize and compress the dynamic range of routing scores.
> > W4: LapSum formula
>
> To clarify the soft top-$k$ selection mechanism, we will add the explicit formulation to Section 3 in the paper. The method identifies a differentiable threshold $x$ by solving the equation:
> $$\mathrm{LapSum}(x) = \sum_{i=1}^n F_{\text{Lap}}(r_i - x) = k,$$
> where $r_i$ are the input scores and $F_{\text{Lap}}$ denotes the Laplace CDF.
> This equation admits a unique solution $\tilde{x}$, which we denote as $\tilde{x} = \mathrm{LapSum}^{-1}(k)$ and is described by an explicit formula. Intuitively, $\mathrm{LapSum}(x)$ acts as a smooth approximation of the counting function $\sum_i \mathbf{1}[r_i \ge x]$, and solving $\mathrm{LapSum}(x)=k$ corresponds to finding a continuous relaxation of the $k$-th order statistic (i.e., the top-$k$ threshold). The resulting soft selection weights are then given by $F_{\text{Lap}}(r_i - \tilde{x})$, which provide a differentiable approximation of the binary top-$k$ mask. This formulation can be viewed as a generalization of the softmax operator: similarly, it produces an output tensor of the same dimensionality as the input. However, unlike softmax (where the outputs form a probability distribution summing to 1), LapSum enforces that the outputs sum exactly to $k$, while each individual element remains in the range $[0, 1]$. This makes it particularly suitable for differentiable top-$k$ selection, where both sparsity and gradient flow are required.

---

> > ### Author Rebuttal · Reviewer_Fdmu · 2026-04-04
> >
> > Thank the authors for the detailed rebuttal — it addresses most of my concerns.
> >
> > I have one remaining suggestion. I understand that truncation is simply ReLU, then after truncation the gradient situation is structurally similar to hard top-k: active experts receive gradient, inactive experts receive zero gradient, and the selection boundary itself is still non-differentiable. An expert just below threshold receives no gradient signal in either case. The difference is that the number of surviving experts becomes input-dependent, but the nature of the gradient flow at the selection boundary is the same.
> > I encourage the authors to include a precise discussion clearly delineating what is and is not differentiable in the full deployed pipeline (SoftTopK + truncation) vs TopK, rather than broadly characterizing it as "differentiable routing." It's more like a routing that enables gradient-based optimization of per-layer expert budgets, i.e. differentiable on k.
> >
> > I'm happy to raise my score if this is clarified.

---

> > > ### Author Response · Authors · 2026-04-04
> > >
> > > Thank you for your thorough evaluation of our paper and for the constructive follow-up. We are glad that the rebuttal addressed most of your concerns. Regarding the remaining point, we agree with your characterization. To precisely delineate differentiability in the deployed pipeline, we will include the following paragraph in the revised manuscript:
> > >
> > > *The truncation operator $\mathcal{T}(z) = z\mathbb{I}_{z>\tau}$ is equivalent to a shifted ReLU: active experts receive gradient through their routing weights, while inactive experts receive none. At the decision boundary, expert selection is non-differentiable, analogous to hard top-$k$ routing. That said, the model remains differentiable with respect to the selection parameter $k$ via the LapSum inverse. This gradient pathway is unavailable when $k$ is a fixed integer, and is what enables learning per-layer expert allocation under the global budget constraint. Moreover, because truncation applies a threshold rather than selecting by rank, the number of active experts is input-dependent, allowing the model to adjust per-token compute. In summary, SoftMoE is a routing mechanism that enables gradient-based optimization of per-layer expert budgets and per-token compute adaptation, rather than a differentiable replacement of hard top-$k$ selection.*

---

### Official Review · Reviewer_wnkB · 2026-03-16

**Soundness:** 2
**Presentation:** 3
**Significance:** 3
**Originality:** 4
**Overall Recommendation:** 4
**Confidence:** 3

**Summary:**

The most common way to train MoEs is via top-k token choice routing, in which k is fixed (each token is always assigned to k experts at every layer). In this work, the authors believe that expert allocation should be dynamic both within a layer and across layers. They propose SoftMoE, which, instead of selecting the top-k experts (where k is fixed), it selects the experts with router weights above a threshold hyperparameter (and if below the threshold, the router score is set to 0, hence not activated). The router is therefore differentiable, and can learn to adaptively allocate compute across layers and tokens. To maintain sparsity, SoftMoE imposes a constraint for a fixed global budget (total K) across layers. Authors train small-scale models and show on some tasks that SoftMoE outperforms standard top-k MoEs.

**Compliance With Llm Reviewing Policy:**

Affirmed.

**Key Questions For Authors:**

On the fairness of evaluation: Are the sparse MoE baselines compute-matched with SoftMoE? Do they take the same number of GPU hours?

Does SoftMoE make it harder to have a balanced load? I.e. Can we see a comparison fo the auxiliary balancing loss curve for SoftMoE vs baseline?

**Limitations:**

Yes

**Strengths And Weaknesses:**

Strengths:

- The method is simple and elegant.

- Paper is well-motivated (adaptive computation for MoEs).

- The analysis on how SoftMoE learns to redistribute compute to the deeper layers is interesting.

Weakensses:

- The models and evaluation benchmarks are too toy, and it’s unclear whether the gains would sustain when scaled up to actually useful scales (>1B active parameter models with >500B tokens)

- Models are extremely undertrained. For example, on the OWT dataset, according to Appendix B, the model is only trained on $2048 \times 240 \times 18000=8B$ tokens.

- Models are very small, with the largest only ~200M active parameters.

- Is the small model size/undertraining the reason why authors chose to use very simple evaluation benchmarks? Would it be possible to see performance on more common pretraining benchmarks for small scales? Namely, MMLU, BBH, MBPP, HumanEval, GSM8K, and MATH.

---

> ### Author Rebuttal · Authors · 2026-03-27
>
> Thank you Reviewer for the careful reading of our paper and constructive feedback. We address each raised concern below.
>
> > Weaknesses points 1 to 3
>
> We conducted experiments within the computational limits available to us and hope to inspire further large-scale investigations. We are strongly convinced that our allgorithm scales well. Our experiments on C4 confirm this, and while we would like to extend the evaluation further, we consistently observe good scalability across all datasets we have tested so far.
>
> > Weaknesses point 4: Is the small model size/undertraining the reason why authors chose to use very simple evaluation benchmarks?
>
> Yes, based on prior literature, we selected benchmarks that are commonly used and appropriate for models of this scale.
>
> > Q1: On the fairness of evaluation: Are the sparse MoE baselines compute-matched with SoftMoE? Do they take the same number of GPU hours?
>
> In the table bellow we report active experts, wall clock training time, time per iteration, throughput and GPU utilization. Throughput and timings in those experiments are affected by compute cluster load, which is not under our control (manifested in GPU utilization). Thats said, the baselines are scaled to be computed matched with SoftMoE taking into account theoritical flops (active experts). Note that except for the number of active experts, all compared models use the same architecture.
>
> | Dataset | Model        | Train-AE | Time(h) | GPUh | Avg Iter Time(ms) | Tokens/s/gpu | GPU Util.(%) |
> | ------- | ------------ | --------:| -------:| ----:| -----------------:| ------------:| ------------:|
> | OWT     | SparseMoE-2  | 2        | 2.2     | 17.7 | 438.96            | 142836       | 89.72        |
> | OWT     | SoftMoE*-1.5 | 1.53     | 2.5     | 20.0 | 495.44            | 126554       | 82.75        |
> | OWT     | SoftMoE-1.5  | 1.63     | 1.9     | 15.5 | 384.93            | 162883       | 94.02        |
>
> > Q2:  Does SoftMoE make it harder to have a balanced load? I.e. Can we see a comparison fo the auxiliary balancing loss curve for SoftMoE vs baseline?
>
> In our experiments SoftMoE has no discernible impact on load balancing. Table bellow reports reative variance in expert load (tokens per expert) across our experiments.
>
> | Dataset | Model     |   $k$    |    $\alpha$    | Load variance (relative)|
> | ------- | --------- | ------:| ------:| ------------------------:|
> | OWT     | SparseMoE | 1    | --   | 0.10                     |
> | OWT     | SparseMoE | 2    | --   | 0.05                     |
> | OWT     | SoftMoE*  | 1.5  | 2    | 0.05                     |
> | OWT     | SoftMoE   | 1.5  | 2    | 0.05                     |
> | OWT     | SparseMoE | 3    | --   | 0.04                     |
> | OWT     | SparseMoE | 4    | --   | 0.04                     |
> | OWT     | SoftMoE*  | 1    | 4    | 0.03                     |
> | OWT     | SoftMoE*  | 2    | 2    | 0.07                     |
> | OWT     | SoftMoE   | 2    | 2    | 0.06                     |
> | C4      | SparseMoE | 1    | --   | 0.04                     |
> | C4      | SparseMoE | 2    | --   | 0.04                     |
> | C4      | SoftMoE*  | 1.5  | 2    | 0.04                     |
> | C4      | SparseMoE | 3    | --   | 0.03                     |
> | C4      | SoftMoE   | 1.5  | 2    | 0.04                     |
> | C4      | SparseMoE | 4    | --   | 0.03                     |
> | C4      | SoftMoE*  | 1    | 4    | 0.02                     |
> | C4      | SoftMoE*  | 2    | 2    | 0.03                     |
> | C4      | SoftMoE   | 2    | 2    | 0.05                     |

---

> > ### Author Rebuttal · Reviewer_wnkB · 2026-04-02
> >
> > Dear authors,
> >
> > Thank you for your carefully written rebuttal and for taking the time to provide additional results.
> >
> > I'm happy with the responses on Q1 and Q2. I also understand that many researchers do not have the compute budget to pretrain models on a larger scale, however, I think having >1B active param experiments on reasonable token horoizon would make the paper a lot stronger, as this is often the smallest "useful" model to have non-random base model performance on less toy tasks such as MMLU/math tasks /coding tasks. Therefore, I would like to maintain my score of weak accept.

---

> > > ### Author Response · Authors · 2026-04-03
> > >
> > > We appreciate that the additional results were sufficient to resolve the fairness of evaluation and load balancing questions. Thank you for your valuable feedback and overall assessment of our paper.

---

### Decision · Program_Chairs · 2026-04-30

**Decision:**

Accept (regular)

**Comment:**

This paper proposes a routing mechanism for MoE models that replaces the hard top-k routing with a truncated LapSum-based selection method. REviewers find the method well motivated, simple and practical. The observation that SoftMoE learns non-uniform expert allocation favoring deeper layers is an interesting empirical result.

Concerns primarily revolve around scope of experiments, and questions around reults on larger-scale settings. Other questions around fairness, load balancing etc. seem to have been addressed by the rebuttal.

Overall, all reviewers vote in favor of acceptance (either as a weak accept or accept).